DNA barcoding continues to identify endangered species of shark sold as food in a globally significant shark fin trade hub

Selena Shen Kai-Lin 1
Cheow Jin Jie 1
Cheung Abigail Belle 1
Koh Ryan Jia Rong 1
Koh Xiao Mun Amanda 1
Lee Yun Ning 1
Lim Yan Zhen 1
Namatame Maya 1
Peng Eileen 1 2
Vintenbakh Vladislav 1
Lim Elisa X.Y. 1
Wainwright Benjamin John 1 3 Ben.Wainwright@Yale-NUS.edu.sg
1 Yale-NUS College, National University of Singapore , Singapore
2 Yale University , New Haven , CT , USA
3 Department of Biological Sciences, National University of Singapore , Singapore
Doğdu Servet
Electronic publication date: 2024 Jan 3
Publication date: 2024
Volume: 12
Electronic Location ID: e16647
Received 2023 Sep 21; Accepted 2023 Nov 20
Copyright: © 2024 Selena Shen et al.
Copyright year: 2024
Copyright holder: Selena Shen et al.
License: This is an open access article distributed under the terms of the Creative Commons Attribution License, which permits unrestricted use, distribution, reproduction and adaptation in any medium and for any purpose provided that it is properly attributed. For attribution, the original author(s), title, publication source (PeerJ) and either DOI or URL of the article must be cited.
License URL: https://creativecommons.org/licenses/by/4.0/

Keywords: CITES, Conservation, IUCN, Mislabelling, Singapore, Seafood

Funding: This research did not receive any specific grant from funding agencies in the public, commercial, or not-for-profit sectors.

==============================
Shark fins are a delicacy consumed throughout Southeast Asia. The life history characteristics of sharks and the challenges associated with regulating fisheries and the fin trade make sharks particularly susceptible to overfishing. Here, we used DNA barcoding techniques to investigate the composition of the shark fin trade in Singapore, a globally significant trade hub. We collected 505 shark fin samples from 25 different local seafood and Traditional Chinese Medicine shops. From this, we identified 27 species of shark, three species are listed as Critically Endangered, four as Endangered and ten as Vulnerable by the International Union for Conservation of Nature (IUCN). Six species are listed on CITES Appendix II, meaning that trade must be controlled in order to avoid utilization incompatible with their survival. All dried fins collected in this study were sold under the generic term “shark fin”; this vague labelling prevents accurate monitoring of the species involved in the trade, the effective implementation of policy and conservation strategy, and could unwittingly expose consumers to unsafe concentrations of toxic metals. The top five most frequently encountered species in this study are Rhizoprionodon acutus, Carcharhinus falciformis, Galeorhinus galeus, Sphyrna lewini and Sphyrna zygaena. Accurate labelling that indicates the species of shark that a fin came from, along with details of where it was caught, allows consumers to make an informed choice on the products they are consuming. Doing this could facilitate the avoidance of species that are endangered, and similarly the consumer can choose not to purchase species that are documented to contain elevated concentrations of toxic metals.

Introduction

The shark fin trade is global in nature and is supplied by fisheries across the world’s oceans; this trade is in part responsible for the large declines in shark populations. Two-thirds of the sharks involved in the global fin trade are at risk of extinction or come from populations that are in decline (Cardeñosa et al., 2022; Clark-Shen et al., 2023; Sherman et al., 2023). Shark fins within this trade are commonly exported in dried forms and sold under generic terms (e.g., shark fin or dried seafood) rather than detailing the species of origin. This ambiguous or deliberately vague labelling makes enforcement and monitoring of the trade challenging (Cardeñosa et al., 2017).

Despite a growing awareness of the need to conserve sharks, the practice of consuming shark fin products for celebratory or health reasons remains common throughout much of Asia (Clarke, Milner-Gulland & Bjørndal, 2007; Dent & Clarke, 2015; Teo, 2015; Ip et al., 2021; Choy & Wainwright, 2022). This consumption supports a near USD 1 billion industry (Worm et al., 2013; Dent & Clarke, 2015) that contributes to the increasing extinction risk that many species of shark now face (Sherman et al., 2022; Dulvy et al., 2021). As a consequence of their life history characteristics (e.g., their slow growth rates, late sexual maturity, and low reproductive output) (Cardeñosa et al., 2018; Frisk, Miller & Fogarty, 2001) sharks are particularly susceptible to the pressures associated with overfishing. Their removal from marine ecosystems can disrupt ecological communities through the selective removal of upper trophic level predators, resulting in trophic cascades that have the potential to disrupt ecosystem stability (Bascompte, Melián & Sala, 2005).

Efforts to regulate unsustainable fishing include the setting of sustainable catch quotas and the implementation of rules through the regulation of trade. These efforts can be implemented under frameworks such as the Convention on International Trade in Endangered Species of Wild Fauna and Flora (CITES), or the International Union for the Conservation of Nature (IUCN) (Dulvy et al., 2021). However, the enforcement and effectiveness of these regulations is severely restricted by the process of shark finning which removes diagnostic characteristics (Shivji et al., 2002; Bornatowski, Braga & Vituleb, 2014), making accurate species level identifications nearly impossible. Mislabelling, or the deliberately vague labelling of products to conceal the species of origin is a common practice throughout the global seafood trade (Marko, Nance & van den Hurk., 2014; Chang et al., 2021; French & Wainwright, 2022; Neo, Kibat & Wainwright, 2022). This not only makes successful shark conservation and effective policy creation challenging, it can also expose consumers to potentially unsafe concentrations of toxic metals (Marko, Nance & van den Hurk., 2014; Chan et al., 2023).

As apex predators, sharks are particularly vulnerable to the biomagnification of toxins (Tiktak et al., 2020) and different species of shark accumulate these toxins at different rates. Pelagic species, or those feeding at depths of 1,000 m or more, are expected to contain elevated levels of mercury in comparison to those that are restricted to coastal areas (Choy et al., 2009; Kojadinovic et al., 2006; Tiktak et al., 2020). Similarly, some elasmobranch species have been reported to contain concentrations of arsenic that exceed recommended safe consumption limits by more than 20 times (Whitcraft, O’Malley & Hilton, 2014). Considering the species-specific differences that exist in how readily toxic metals can be accumulated and the negative consequences that exposure can have on human health (e.g., central nervous system and brain damage, hypertension and coronary heart disease) (Pacyna et al., 2010; Houston, 2011), it would be prudent for governments to implement a unilateral labelling scheme that clearly identifies the species of origin and the location of capture. Doing this will allow consumers to make an informed choice and choose only to eat shark fins that come from populations that are sustainably managed, from species that are not endangered, or from species that are documented to accumulate toxins less rapidly and at lower concentrations (Rodriguez-Mendivil et al., 2019; Wang et al., 2022; Riesgo et al., 2023). This is an important consideration; work examining the toxic metal concentrations of fins collected in Singapore reports numerous instances where concentrations are above established safe consumption limits, with significant differences in toxic metal concentrations observed between species, and between species that inhabit coastal or pelagic environments (Chan et al., 2023).

Even for experts, making accurate species identifications from dried fins can be challenging and is frequently impossible, because of this, DNA barcoding techniques have been developed to aid in species identifications where this is not possible by visual methods alone. These techniques rely upon species-specific nucleotide differences within a given gene (e.g., mitochondrial COI) to make accurate identifications. The accurate identification of the species involved within the fin trade is essential when attempting to determine sustainable catch quotas, evaluation of IUCN and CITES designations and the implementation conservation management plans (Wainwright et al., 2018). Here, we use DNA barcoding techniques to make species level identifications of fins collected in the retail markets of Singapore, a globally significant trade hub and the world’s second-largest importer and re-exporter of shark fin in terms of value (Boon, 2017). We collected samples from a variety of stores including Traditional Chinese Medicine (TCM) shops, supermarkets and seafood retailers. Similar to other work performed in Singapore (Wainwright et al., 2018) and the region (Seah et al., 2022), we hypothesise that this work will find numerous species of endangered sharks throughout all of our collected samples.

While the application of DNA barcoding to identify the species of shark that a fin came from is not new, it remains important that regular monitoring takes place, especially as the composition of sharks within the trade is not static within a country, over time or between countries. The differences observed between countries and time points are likely indicative of the variety of markets and global nature of the fisheries that supply trade hubs (Drescher et al., 2022). Without repeated monitoring, it is impossible to understand the impact of new regulations and policy and how they could affect the species that are caught. From a conservation and policy perspective, knowing the species involved in the trade is vital information that can be considered when updating catch quotas, or revising the conservation status of specific species. In the context of public health, knowing the species of shark that a fin came from allows the avoidance of those species that can accumulate heavy metals at an elevated rate.

Methods

Sample collection

In January 2023, we collected 505 shark fins across Singapore. Our collection method followed that of Drescher et al. (2022). Briefly, a list of all current shark fin retailers in Singapore was compiled and 25 shops to visit and purchase fins from were chosen at random. At each shop, the proprietor haphazardly selected a minimum of 20 fins from large containers that held a mixture of fins. Purchased fins were not placed in any preservative, this was deemed unnecessary as the fins were displayed and stored in the shops at room temperature. All DNA extractions were completed within 4 days of sample purchase.

DNA extraction and PCR amplification

DNA was extracted from a 15–25 mg piece of fin. To minimise the possibility of contamination that could arise because of storage in large mixed containers of fins, all fin samples for DNA extraction were taken from an internal part where they are less likely to encounter tissue or debris from other fins, and all tools were sterilised between samples. DNA extraction was performed with the Qiagen DNeasy® Blood and Tissue Kit (Qiagen, Hilden, Germany), and the extraction process followed the manufacturer’s instructions with the slight modification that DNA was eluted in 50 μL of elution buffer.

Due to the degraded and processed nature of the fins used in this study, we opted to amplify a reduced portion of the mitochondrial COI gene with the following primers. Forward primer mlCOIintF (5′-GGW ACW GGW TGA ACW GTW TAY CCY CC-3′) (Leray et al., 2013) and reverse primer LoboR1 (5′-TAA ACY TCW GGR TGW CCR AAR AAY CA-3′) (Lobo et al., 2013) were used in PCR to amplify a ~350 bp fragment of mitochondrial DNA. Each reaction was conducted in a 25 μL volume containing: 1 μL each of the forward primer and the reverse primer, both at 10 mM, 1 μL bovine serum albumin (BSA), 7.5 μL of nuclease-free water, 12.5 μL GoTaq mastermix green, and 2 μL of undiluted DNA template. The thermal cycling profile for amplification consisted of: 5 repeats of 94 °C for 30 s, 48 °C for 2 min, 72 °C for 1 min, then 35 repeats of 94 °C for 30 s, 54 °C for 2 min, 72 °C for 1 min, then 72 °C for 5 min (Wainwright et al., 2018). The initial five repeats of our cycling protocol dramatically improves the success of the PCR. To confirm successful DNA amplification prior to sequencing, samples were run on a 1% agarose gel. Bidirectional Sanger sequencing and enzymatic cleaning was performed by Bio-Basic Asia Inc. using an ABI 3730xl DNA Analyser. Sequences were visualised with Geneious Prime (v2023.0.4) (Kearse et al., 2012).

Sequence identification

High quality sequences (i.e., only those with well-defined peaks, and no ambiguous base calls) were then referenced against the Barcode of Life Data System (BOLD) and GenBank. The criteria for positive species identification was: (1) BOLD returning a 100% match for a single species, and (2) an identical top match in BOLD and GenBank (Neo, Kibat & Wainwright, 2022; Choy & Wainwright, 2022). For any sequence that could not be identified at the species level, we deferred to the top matching genus returned in both databases. After quality control, which involved removing primer sequences and any low quality bases at the beginning or end of our sequences, all sequences used in species identification were between 200–350 bp in length.

Results

Overall, we successfully identified 378 of the 505 samples at either the genus or the species level. We positively identified 27 shark species across 16 genera. Six identified shark species are listed on CITES Appendices II (2022) (n = 105) (Table 1 and Fig. 1). The IUCN considers 17 of the 27 species threatened, of which three species are listed as Critically Endangered (n = 65), four as Endangered (n = 13) and ten as Vulnerable (n = 136). The remaining eight species were listed as Near Threatened by the IUCN (n = 35) (Table 1 and Fig. 1).

Table 1 Species identification, common names, occurrence, IUCN Red List status and CITES status.

Scientific name	Common name	Occurrence	Occurrence %	IUCN status	CITES	
Rhizoprionodon acutus	Milk shark	40	10.6	VU	–	
Carcharhinus falciformis	Silky shark	39	10.3	VU	II	
Galeorhinus galeus	School shark	34	9.0	CR	–	
Sphyrna lewini	Scalloped hammerhead	30	7.9	CR	II	
Sphyrna zygaena	Smooth hammerhead	25	6.6	VU	II	
Rhizoprionodon oligolinx	Grey sharpnose shark	12	3.2	NT	–	
Carcharhinus brevipinna	Spinner shark	9	2.4	VU	–	
Carcharhinus sorrah	Spot-tail shark	8	2.1	NT	–	
Isurus oxyrinchus	Shortfin mako shark	8	2.1	EN	II	
Carcharhinus leucas	Bull shark	7	1.9	VU	–	
Carcharhinus melanopterus	Blacktip reef shark	5	1.3	VU	–	
Galeocerdo cuvier	Tiger shark	4	1.1	NT	–	
Hemipristis elongata	Snaggletooth shark	4	1.1	VU	–	
Triaenodon obesus	Whitetip reef shark	4	1.1	VU	–	
Carcharhinus macloti	Hardnose shark	3	0.8	NT	–	
Loxodon macrorhinus	Sliteye shark	3	0.8	NT	–	
Mustelus canis	Dusky smooth-hound	3	0.8	NT	–	
Eusphyra blochii	Winghead shark	2	0.5	EN	–	
Hemigaleus microstoma	Sicklefin weasel shark	2	0.5	VU	–	
Alopias pelagicus	Pelagic thresher	2	0.5	EN	II	
Chaenogaleus macrostoma	Hooktooth shark	1	0.3	VU	–	
Lamiopsis temminckii	Broadfin shark	1	0.3	EN	–	
Hemigaleus australiensis	Australian weasel shark	1	0.3	LC	–	
Chiloscyllium punctatum	Brownbanded bamboo shark	1	0.3	NT	–	
Prionace glauca	Blue shark	1	0.3	NT	–	
Rhizoprionodon taylori	Australian sharpnose shark	1	0.3	LC	–	
Sphyrna mokarran	Great hammerhead	1	0.3	CR	II	
Carcharhinus spp.	N/A	79	20.9	N/A	N/A	
Mustelus spp.	N/A	47	12.4	N/A	N/A	
Rhizoprionodon spp.	N/A	1	0.3	N/A	N/A	
Note:

CR, Critically Endangered; EN, Endangered; VU, Vulnerable; NT, Near Threatened; LC, Least Concern.

Figure 1 Bar plot showing the species identified, their occurrence and coloured by IUCN status.

An asterisk (*) indicates CITES appendix II listed species.

We were unable to identify 127 fins past the level of genus. Of these, 79 fins came from the genus Carcharhinus, 47 from the genus Mustelus and one came from the genus Rhizoprionodon (Table 1 and Fig. 1). Out of the top five identified species, three are listed as Vulnerable by the IUCN and two are listed as Critically Endangered. In total, these five species accounted for 168 out of 505 samples (33% of total samples). The most commonly identified species was Rhizoprionodon acutus (n = 40) which is listed as a Vulnerable species on the IUCN Red list. This was closely followed by Carcharhinus falciformis (n = 39) which is listed as Vulnerable and Galeorhinus galeus (n = 34) which is listed as Critically Endangered.

Discussion

We successfully identified 378 samples uncovering 27 different shark species from 16 genera. Consistent with expectations and in agreement with previous work examining the shark fin trade (Holmes, Steinke & Ward, 2009; Marchetti et al., 2020), many of these species are listed as threatened, or have a degree of control imposed upon their trade. Of the 27 species we identified, 17 are listed as threatened (Critically, Endangered or Vulnerable) on the IUCN Red list, with six listed on CITES Appendices II (2022) (Table 1 and Fig. 1).

Rhizoprionodon acutus, commonly known as the milk shark, was the most frequently encountered species in the present study (n = 40). This species is listed as Vulnerable on the IUCN Red List, and is not presently listed on any CITES appendices. R. acutus is frequently encountered in DNA barcoding research performed throughout Asia (Cardeñosa et al., 2020; Fields et al., 2017; Liu et al., 2021). On account of its long reproductive cycle that has a yearlong gestation period (Olsen, 1954) and its slow growth rate (Lucifora, Menni & Escalante, 2004), this species is susceptible to overexploitation and could warrant a higher degree of protection, especially as estimates suggest that populations have declined by 30% (Australian Government Shark Report, 2019).

Carcharhinus falciformis, the silky shark, was the second most commonly encountered species in the current work (n = 39). This species is consistently one of the most frequently encountered in the Singapore fin retail trade (Wainwright et al., 2018; Drescher et al., 2022), and within the markets of Indonesia, Hong Kong, Mainland China and Malaysia (Sembiring et al., 2015; Cardeñosa et al., 2018; Fields et al., 2017; Seah et al., 2022). Similar to Rhizoprionodon acutus, C. falciformis is frequently encountered as bycatch in tuna fisheries (Poisson et al., 2014; Curnick et al., 2020; Francis et al., 2023), and as with other sharks its life history characteristics make it vulnerable to overexploitation.

Galeorhinus galeus, the school shark, is the third most common species in this work (n = 34). While this species has been recorded in previous work performed in Singapore (Liu et al., 2021; Drescher et al., 2022) and throughout the Asia-Pacific region (Smith & Benson, 2001; Fields et al., 2015; Fields et al., 2017), this is the first time it has been encountered at such high abundance in surveys performed within Singapore. This species is listed as critically endangered by the IUCN, but is not currently listed on the CITES appendices. G. galeus is an important species in fisheries, where it is reported to be one of the most extensively fished sharks in the world with sizable fisheries in South America, California and Southern Australia (FAO, 2023). G. galeus has been documented as having one of the lowest intrinsic rebound potentials of all sharks assessed by Smith, Au & Show (1998), with populations from North America and Australia all having experienced significant declines in the 1950s and show no current indications that these populations will rebound (FAO, 2023). In light of the limited rebound potential and the acknowledged large scale fisheries that target this species and the IUCN designation as critically endangered, it is likely that G. galeus would benefit from the trade regulations that a CITES listing would bring.

The fourth and fifth most frequently encountered sharks in this study are Sphyrna lewini, the Scalloped Hammerhead (n = 30) and Sphyrna zygaena, the Smooth Hammerhead (n = 25). S. lewini is another species that is commonly encountered in the fin trade throughout Asia (Sembiring et al., 2015; Liu et al., 2021; Seah et al., 2022), and this species is one of the top four most commonly traded sharks on the international market (Fields et al., 2017). It is listed as critically endangered by the IUCN and trade of this species is controlled by its inclusion on CITES Appendices II (2022). Despite these designations, and as this current work along with previous studies show, S. lewini is still readily traded on a global scale, likely at a level that is incompatible with its continued survival. S. zygaena is another species that is more prevalent in the current study than previous work; it is a semi-pelagic species that is also vulnerable to becoming bycatch in tuna fisheries, although it is less frequently encountered as bycatch in comparison to others (Santos & Coelho, 2018).

R. acutus, C. falciformis, G. galeus, S. lewini and S. zygaena are the top five most frequently encountered species in this work. All five are designated as threatened by the IUCN (Critically endangered (n = 2) and Vulnerable (n = 3)) and three are listed on CITES Appendices II (2022). Considering the generic ‘shark fin’ label that all these fins were sold under, it is highly probable that there is no record of these fins entering Singapore. Information on what species are traded is critical when determining management strategies and appropriate catch quotas (Holmes, Steinke & Ward, 2009). Without the application of DNA barcoding techniques, it is unlikely the full extent of trade in these species would come to light.

Despite the high fishing pressure that Prionace glauca, the blue shark, is exposed to and its prevalence in the Hong Kong fin trade as one of the most frequently encountered species (Fields et al., 2017), we only detected one occurrence in this work. Using similar methods to those used here, P. glauca was previously observed much more frequently in the Singapore trade between 2018 & 2021 (Wainwright et al., 2018; Liu et al., 2021), yet it is only found once in this study and was completely absent in one performed in 2022 (Drescher et al., 2022). Prionace glauca is already heavily exploited (Simpfendorfer & Dulvy, 2017) and the limited occurrence of this species in ongoing work is likely further evidence of the unsustainable fishing pressure that has removed, and continues to remove sharks from the ocean.

As with other studies employing DNA mini-barcoding approaches (Fields et al., 2017; Liu et al., 2021), we were unable to resolve a number of samples within the genus Carcharhinus to the species level. This is a consequence of the limited genetic differentiation that exists between species within this genus (Ovenden et al., 2010). Due to the nature of the samples collected (i.e., dried and processed fins) and the consequent degradation of DNA, mini-barcoding approaches are necessary to achieve PCR amplification. As a result of the reduced resolution of these approaches, we were unable to identify 79 samples beyond the Carcharhinus genus level. Other approaches using different genes (e.g., NADH2) have been used to further distinguish closely related shark species (Spaet & Berumen, 2015; Marchetti et al., 2020), but the length of this gene and the degraded nature of our samples mean it is not suitable for use in work such as ours.

Knowing what species of shark a fin came from is important from a human health perspective. As apex predators, sharks can accumulate significant concentrations of toxic metals in their tissues, a consequence of biomagnification and the high trophic levels they occupy (O’Bryhim et al., 2017; Ong & Gan, 2017; Boldrocchi et al., 2019; Álvaro-Berlanga et al., 2021). Some species of shark and shark populations are more susceptible to this accumulation than others (Glover, 1979; Rodriguez-Mendivil et al., 2019; Wang et al., 2022; Riesgo et al., 2023). For example, R. acutus, the most frequently encountered species in this work, has been shown to accumulate toxic metals such as selenium, mercury and other trace metals to concentrations that can be potentially harmful to humans (Ong & Gan, 2017; Rodríguez-Gutiérrez et al., 2020; Boldrocchi et al., 2021). The same is true of G. galeus and other species of sharks (Santos & Coelho, 2018). In fact, research performed in Singapore shows significant differences exist in the concentrations of toxic metals found in fins between species, and between pelagic and oceanic dwelling species (Chan et al., 2023). If the species of origin is not indicated, DNA barcoding is a method that can be used to make these identifications and help guide consumer choice. Just as many nations have adopted ‘traffic light’ coloured health warnings to effectively inform consumers of the sugar or salt content of their food, we envisage a similar system that warns consumers of their potential exposure to mercury or other toxic metals based upon the species that a shark fin came from. This relies on a concerted effort by policy and enforcement agencies to ensure that the species of origin is clearly marked on each fin, something we acknowledge that is not easy, but is not impossible. For example, the European Union mandates strict labelling standards that detail the species and its origin so that provenance can be determined throughout supply and processing chains (Paolacci et al., 2021). As DNA barcoding, sequencing and molecular identification techniques improve along with increases in sequencing capacity, accuracy and reductions in turnaround times, it becomes increasingly feasible that techniques such as this can be deployed at the point of entry to determine species IDs. For example, it is now possible to make accurate IDs within 60 min with only minimal equipment (But et al., 2020).

The identification of the shark species that a fin came from equips consumers with the awareness and autonomy to make informed purchases, allowing for the avoidance of fins from species that are known to be overfished or from shark species that are suspected to have high concentrations of potent neurotoxins (Melián & Bascompte, 2004; Nowicki et al., 2021). Informed consumption takes on additional importance in vulnerable demographics, such as elderly populations who are more likely to be medically predisposed and sensitive to the adverse effects of excessive mercury and toxic metal consumption. This is particularly relevant in the case of shark fin consumption; it is the older generation who are much more likely to consume shark products (Yeo, 2022) and be exposed to toxic metals.

DNA barcoding is now a routine technique used throughout the world in a variety of situations and for a range of purposes; however, there is still much value in studies such as this to understand the composition of the trade at a given point in time. Without this work and its associated molecular identifications, it is very likely that many of these fins would remain unidentified, and the extent to which various species of shark involved in the shark trade would remain unknown.

Supplemental Information

Supplemental Information 1 Supplemental File DNA Sequences.

Examples of DNA sequences used in this work. Note for each species, only a single haplotype was found in all instances.

Click here for additional data file.

Additional Information and Declarations

Competing Interests

Author Contributions

Data Availability

The authors declare no conflict of interest.

Kai-Lin Selena Shen conceived and designed the experiments, performed the experiments, analyzed the data, prepared figures and/or tables, authored or reviewed drafts of the article, and approved the final draft.

Jin Jie Cheow conceived and designed the experiments, performed the experiments, analyzed the data, prepared figures and/or tables, authored or reviewed drafts of the article, and approved the final draft.

Abigail Belle Cheung conceived and designed the experiments, performed the experiments, analyzed the data, prepared figures and/or tables, authored or reviewed drafts of the article, and approved the final draft.

Ryan Jia Rong Koh conceived and designed the experiments, performed the experiments, analyzed the data, prepared figures and/or tables, authored or reviewed drafts of the article, and approved the final draft.

Amanda Koh Xiao Mun conceived and designed the experiments, performed the experiments, analyzed the data, prepared figures and/or tables, authored or reviewed drafts of the article, and approved the final draft.

Yun Ning Lee conceived and designed the experiments, performed the experiments, analyzed the data, prepared figures and/or tables, authored or reviewed drafts of the article, and approved the final draft.

Yan Zhen Lim conceived and designed the experiments, performed the experiments, analyzed the data, prepared figures and/or tables, authored or reviewed drafts of the article, and approved the final draft.

Maya Namatame conceived and designed the experiments, performed the experiments, analyzed the data, prepared figures and/or tables, authored or reviewed drafts of the article, and approved the final draft.

Eileen Peng conceived and designed the experiments, performed the experiments, analyzed the data, prepared figures and/or tables, authored or reviewed drafts of the article, and approved the final draft.

Vladislav Vintenbakh conceived and designed the experiments, performed the experiments, analyzed the data, prepared figures and/or tables, authored or reviewed drafts of the article, and approved the final draft.

Elisa X. Y. Lim conceived and designed the experiments, performed the experiments, analyzed the data, prepared figures and/or tables, authored or reviewed drafts of the article, and approved the final draft.

Benjamin John Wainwright conceived and designed the experiments, performed the experiments, analyzed the data, prepared figures and/or tables, authored or reviewed drafts of the article, and approved the final draft.

The following information was supplied regarding data availability:

The DNA sequences are available in the Supplemental File.

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
