# Peer review of "DNA barcoding continues to identify endangered species of shark sold as food in a globally significant shark fin trade hub"

_PeerJ, doi:10.7717/peerj.16647_

## Round 0.1 · original submission · Major Revisions

Dear Author,
I find your article useful, but it needs some editing.
I would be grateful if you would accept the Reviewers and my suggestions.

In Introduction
-- In line 50-53, When you provide such information, you should write the market value closest to the current year.
--In the introduction you should mention more about the importance of DNA barcoding and why it should be done.
--Unfortunately, your hypothesis is not clear. Your hypothesis should be more descriptive. It should convey the importance of your study to the reader well.

In Methods
--Explain the sampling in a little more detail.
--You can add a photo from the markets related to sampling as a figure.

Results
--You say that, a total of 512 samples, 378 were identified (512-378=134 not defined) (in line 144). But, in the other sentence, you say that 127 samples are not defined (in line 150). Please correct this situation.
--Please add the results of the comparison with Bold and Genbank databases as a table to the publication.
-- Add the sequences obtained in your study to Genbank or Bold database and add the accession numbers to Table 1.
-- It would be more accurate if you add some genetic data to the results. Such as a phylogenetic tree (as Neighbour Joining tree) for your identified species.

Discussion
--In general, the discussion section needs to be reshaped according to the new hypothesis.
-- You can compare your results with other DNA barcoding studies.
-- Explain why DNA barcoding is important to the shark fins with this study.

Reviewer 1 ·

Basic reporting

The article is well-written in clear, concise English and reports all results meaningfully and clearly. I found it an easy and enjoyable read. While not explicitly hypothesis-based, the research still falls within the scope of the journal, and I think it will make a useful contribution. As the authors note, this is not the first study to DNA-barcode shark fins, but they report new data for Singapore, and frame their results in terms of the conservation and public health implications. I have a few very minor editorial notes (see below) but otherwise I think it warrants publication.

My only comment on the content of the study is that the conservation and health implications discussed by the authors fall largely on individual consumer choice and do not discuss structural or institutional measures that might be taken to improve shark conservation overall. Personal choice can be helpful, but overall changes in the health of wild shark populations will not change unless there are larger-scale measures imposed. I don’t think it should hold back publication but philosophically, I’d like to see more treatment of how/what policy might be enacted in the region.

Here are the small notes I had:
Line 57: “upper-level trophic predators” should be “upper trophic level predators”
Lines 193 & 199: “critically threatened” should be “critically endangered”
Line 234: Start a new sentence after “perspective”.

Experimental design

The study is clearly defined and carried out as planned.

Validity of the findings

Findings are valid and clearly verified in the context of the data collected.

Reviewer 2 ·

Basic reporting

No Comments

Experimental design

No Comments

Validity of the findings

No Comments

Additional comments

Overall, it a good manuscript which meets all standards to publish in this journal. However, I made some detailed comments regarding some aspect in your manuscript. Please look at all comments which I made regarding the original manuscript and provide your feedback or clarifications regarding my comments.

Annotated reviews are not available for download in order to protect the identity of reviewers who chose to remain anonymous.

Reviewer 3 ·

Basic reporting

No comment

Experimental design

No comment

Validity of the findings

No comment

Additional comments

The manuscript titled "Identification of Endangered Shark Species in Global Shark Fin Trade Hub Using DNA Barcoding" details the authors' application of DNA barcoding analysis to differentiate shark species through fin samples. The study's analyses are meticulously executed, with no apparent flaws in the methodology. However, my critique of the manuscript revolves around its lack of originality. The authors themselves acknowledge that similar research has been undertaken before (Wainwright et al., 2018).

Aside from this, I have only minor comments:
L82 Please change “Doing this allows” to “Doing this will allow”
L112- Please rephrase the sentence
L130 Mention the concentration of DNA (ng/ul)
L130-131: Please mention the reason why initial five cycles were run on different thermal cycling
parameters
L133: Please mention the instrument model
L146: Refer to Table 1; Figure 1
L151: Refer to Table 1; Figure 1
L200: Delete “Yet”

---

## Round 0.2 · Minor Revisions

Dear Author,

Thank you for accepting and implementing most of the suggested revisions.
We believe that your article has become better with the changes you have made.
We kindly ask you to consider and revise the attached Reviewer 2 comments.

Reviewer 1 ·

Basic reporting

The article is well-reported and falls within the journal's guidelines. I favored publication upon initial submission, and the revised manuscript has been improved through the minor revisions suggested by the other reviewers. I still favor publication.

Experimental design

no comment

Validity of the findings

The findings remain valid.

Additional comments

The guidelines of PeerJ were developed to enable publication of exactly this sort of study, where the reported results are of use to the scientific community but may not be novel or specifically hypothesis-driven and the authors address this in their manuscript.

I believe the authors responded appropriately to the editor's and reviewers' critiques in their rebuttal. The revised manuscript is a useful contribution and warrants publication.

Reviewer 2 ·

Basic reporting

Line 43: Could you clarify this phrase? Does it refer to naming something? Please correct or clarify the sentence.

Line 43: It should be “;” instead of “,”

Line 48: Please specifically state the generic terms or labels they used in parentheses “XXX”’: it would then say like: … and sold under generic terms “Shark Fins” rather than …

Line 146: Once I counted the lengths of all sequences one by one in the SI section, the actual lengths ranged from 229 bp to 318 bp, which is less than 350 bp. Why did you state ~350 bp? Do you mean ~350 bp will include the primer set? To provide a more accurate statement, I suggest it says, “… to amplify a fragment of mitochondrial DNA in the range of 200–350 bp.”

Line 265: Please clarify this phrase or correct the sentence. Perhaps it was just missing a comma.

Experimental design

No comments

Validity of the findings

No comments

Additional comments

Thanks for responding with some feedback on the first review. Here, I provide some final suggestions and comments for the manuscript to get published afterward, so please look at them.

Annotated reviews are not available for download in order to protect the identity of reviewers who chose to remain anonymous.

Reviewer 3 ·

Basic reporting

NA

Experimental design

NA

Validity of the findings

Novelty of the manuscript is lacking.

Additional comments

The manuscript has been revised according to all the recommended changes by the authors, resulting in a significant enhancement in the paper's quality. I am now confident that the manuscript is suitable for publication in PeerJ.

---

## Round 0.3 · accepted · Accept

Dear Author,
We have reached a decision regarding your submission to Peer J "DNA barcoding continues to identify endangered species of shark sold as food in a globally significant shark fin trade hub".

Our decision is to: Accept the Submission

Congratulations and thank you for considering Peer J as a publication outlet for your work.

Sincerely,
Dr. Servet A. Doğdu